# Parametric and Experimental Modeling of Axial-Type Piezoelectric Energy Generator with Active Base

Alexander V. Cherpakov [1,2,*], Ivan A. Parinov [1] and Rakesh Kumar Haldkar [1]

1   I. I. Vorovich Mathematics, Mechanics and Computer Sciences Institute, Southern Federal University, 344090 Rostov-on-Don, Russia; parinov_ia@mail.ru (I.A.P.); rakeshhaldkar@gmail.com (R.K.H.)
2   Department of Information Systems in Construction, Don State Technical University, 344000 Rostov-on-Don, Russia
*   Correspondence: alex837@yandex.ru

**Abstract:** A computational and experimental approach to modeling oscillations of a new axial-type piezoelectric generator (PEG) with an attached mass and an active base is considered. A pair of cylindrical piezoelements located along the generator axis is used as an active base. Plate-type piezoelectric elements, made in the form of two bimorphs on an elastic PEG base, use the potential energy of PEG bending vibrations. Energy generation in cylindrical piezoelectric elements occurs due to the transfer of compressive forces to the piezoelectric element at the base of the PEG during excitation of structural vibrations. The active load scheme is selected separately for each piezoelectric element. Numerical simulation was performed in the ANSYS FE analysis package. The results of modal and harmonic analysis of vibrations are presented. A technique for experimental analysis of vibrations is presented, and a laboratory test setup is described. Numerical and experimental results are presented for the output characteristics of a piezoelectric generator at a low-frequency load. For one of the versions of the generator and a certain displacement amplitude for a frequency of 39 Hz, in the results of a comparative experimental analysis at a load of 10 kΩ, the maximum output power for each cylindrical piezoelectric element was 2138.9 μW, and for plate-type piezoelectric elements, respectively, 446.9 μW and 423.2 μW.

**Keywords:** piezoelectric generator; axial type; proof mass; finite element modeling; physical experiment; measuring setup

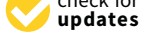



## 1. Introduction

At present, in connection with their development and introduction into production, renewable electrochemical batteries are used to the greatest extent. These electric batteries have the properties of cyclic recovery during a certain resource period and are limited in use when a certain finite operating time is reached [1,2].

Along with renewable batteries, promising devices are used that extract electrical energy from the environment. One of the directions is the extraction of electric energy using solar energy. Some approaches can be presented as examples described in works using solar batteries [3,4]. The energy of the environment in the form of vibration entering compact devices can be extracted using the energy of the movement of air masses (wind), including using energy generation devices in the form of piezoelectric transducers [5,6]. The use of the mechanical energy of the movement of water masses in narrowed volumes, sea tides and tides, and the movement of water masses of rivers and dams, can be represented by the example of [7]. The use of devices for converting thermal energy into electrical energy is presented in [8]. Devices for converting mechanical vibrations of structural elements and soils are presented in the reviews of works [9,10].

Devices for generating electrical energy, the so-called power converters of mechanical energy into electrical energy, using piezoelectric elements, are called piezoelectric generators (PEGs). Their development requires further modernization and improvement. This is

possible only when designing new devices and analyzing their operation under various conditions of dynamic loading. As a review of the current use of PEG shows, such devices require further development due to the low output power and voltage generated.

Basic information about energy generators, as well as problems arising in the development of energy storage devices using piezoelectric elements, were discussed in review articles [11–15]. The primary analysis of review articles shows that models are considered that use piezoelectric elements operating in compression and buckling with tension. Piezoelectric elements can be built into various mechanisms using both pressure and rotational loads. The reviews present the output characteristics of PEG energy of various types of manufacture.

Monographs [16,17] present theoretical works by some authors. Methods for conducting experimental studies are given.

As the literature reviews [12–17] show, the most common element in the generator is a cantilever or a linear type element that is fixed at one or two points. There are schemes with fastening of bending elements in the form of a truss structure. A mass or element in the form of a magnetic attachment is attached to some end or intermediate part of such a device. An electrical load in the form of capacitance C, resistance R or inductance L can be applied to each piezoelectric element separately. Typically, a single-component electrical load is used, which has a certain resistance. In [18], the main topologies of load-relief circuits in the form of electrical energy with the simplest composite circuit of a voltage rectifier in the form of a diode bridge are given. Within the framework of the presented work, it is supposed to measure the voltage at individual electrodes. Thus, under dynamic loading, it is possible to investigate the phase characteristics of the removed potential and the potential on each individual piezoelectric element.

The research in paper [19] proposed a novel hybridization scheme with electromagnetic transduction to enhance the power density of PEHs. The hybrid energy harvester was designed based on the BC–PEH. To compare the power density of the BC–PEH and the hybrid energy harvester, we built a prototype and conducted many experiments. The generator delivers a high voltage of 21.9 V at a drive acceleration of 0.3 g using an array of variable magnets. A peak output power of 103.53 mW is obtained.

In article [20], a cantilever-type generator with an active base was considered. Four cylindrical piezoelements were used in the base. An experimental approach was presented, showing that the maximum output power was 41.8 µW and 7.42 µW for the two types of piezoelectric elements, respectively.

In [21], modeling of the cantilever-type PEG with symmetrical and asymmetrical location of proof mass was presented, and the linear theory of elasticity and electrodynamics was used, taking into account the dissipation of energy as well as the equations of motion in the acoustic approximation. The built FE model was numerically realized in the ANSYS software.

Reference [22] presented a simulation of two generators having a modification with mass elements symmetrically attached, with respect to the main axis of the generator. It was shown that the maximum output power was 6.97 mW.

In [23], FE modeling of a two-axis PEG is considered. The piezoelectric elements were fixed in the form of a bimorph on two cantilevers at the base of the PEG. The calculated output power was 720 µW.

The research in [24] considered a cantilever-type PEG having an aluminum base, with piezoelectric elements in the pinched area and an attached mass at the end. The peculiarity of this generator is that there are stopper devices as a component of the device. The authors fixed the movements of the generator base bar. Analytical modeling of the generator was given, and the results of the nonlinear analysis of the output energy characteristics are presented. The output power of the generator was obtained at a level of up to 4.95 mW. Frequencies were considered up to 40 Hz.

The article [25] considered a broadband PEG with several degrees of freedom based on five resonant frequencies that were fairly close to each other. The PEG consisted of

five end masses, two U-shaped cantilever beams and a straight beam. The selection of resonant frequencies was realized due to the special design of the parameters. The electrical characteristics of the PEG were analyzed through simulation and experiment, confirming that the PEG can effectively expand the operating bandwidth and collect vibration energy at low frequency. Experimental results showed that PEG has five low-frequency resonant frequencies, which are 13, 15, 18, 21 and 24 Hz; under the action of acceleration of 0.5 g, the maximum output power is 52.2, 49.4, 61.3, 39, 2 and 32.1 μW,

The article [26] considered micro-MEMS PEG, which has a wide frequency band with included stoppers; one and two sides are carefully studied. The results of the experiment showed that the operating band is extended to 18 Hz (30–48 Hz) and the corresponding optimal power ranges from 34 to 100 nW, with a base acceleration of 0.6 g. Mathematical modeling based on the application of differential equations of motion was carried out.

In [27], mathematical modeling of PEG was carried out based on the application of Lagrange's electromechanical coupling equations. A PEG prototype with two internal single arms was designed, manufactured and experimentally tested. The accuracy of the proposed mathematical modeling was verified by finite element modeling and experimental results. When tested with low harmonic amplitude, the PEG generated 2.48V, 6.21V and 1.55V at three resonant frequencies between 15 and 30Hz, respectively.

A brief analysis of the modifications and studies of PEG shows the following. The authors of the article [28] optimized the power generator and the software for the analysis of FE, performed using the software ANSYS and ACELAN. The optimal design was based on matching the resonant frequency of the device with the excitation frequency of the environment [28]. In the article [29], a packet-type piezoelectric energy generator was studied. An experimental setup was used and measurements of the response of a multilayer piezoelectric stack in an energy harvester were described. Paper [30] proposed a single-degree fractal structure system for energy collection. The authors optimized the design and experimentally evaluated the performance of the system.

In [31], a piezoelectric power generator was developed with rotation amplitude limitation to avoid resonance conditions. The radially entrained magnetic force was used to collect energy. In [32], the authors considered and analyzed the collection of piezoelectric energy from the characteristics of compliant mechanisms. The authors divided configurations into monostable, multistable, multiple degrees of freedom, frequency upconversion and under load optimization. The authors also introduced a normalized power density to compare the power generation capabilities of a power generator [32].

In [33], the authors designed an energy harvester to extract energy from a smart road. The authors investigated the number of stacks layers, influences of connection mode, number of units and ratio of height to cross-sectional area [33]. Few studies have considered, depending on the area of application, a different design for the piezoelectric generator, in which a direct piezoelectric effect is used when the excitation in the sensitive element vibrates longitudinally (d33), by bending (d31) and by sharing [34–36]. In [37], a three-dimensional finite element analysis is presented for a cantilever plate structure excited by piezoelectric drive sections. The paper considered the modeling of actuators of the optimal configuration of actuators for selective excitation of the modes of a cantilever plate structure. Such elements can be used for technical analysis of vibrations of various structures using MEMS technologies [38].

The aim of this research is to carry out computational and experimental studies for estimation of the output parameters of a new type of piezoelectric energy generator of axial type, which has bimorph active structures on the base bar and symmetrically fixed piezoelectric cylinders at the base, located co-axially to bar; this is a new design of axial type piezoelectric generator. This design harvests energy from $d_{31}$ and $d_{33}$ simultaneously. In this design, proof mass can be mounted on the duralumin beam in between piezoelectric patches and the screw side. This variation in fixing the proof mass endows flexibility onto the natural frequencies of the PEGs. As per the mechanical vibration input, the first natural frequency can be adjected within the limit for high power output. This design provides

flexibility and enhances the output power options compared to the other previous designs. The mechanical vibrations are used as an input parameter. This study concerns the new design of a generator, in which proof mass plays a key role in achieving higher power output. The analysis is also carried out on the electrical load dependency.

The paper is organized as follows: Section 2 provides a description of model parameters and the electric scheme presents a description of axial-type PEG elements, a schematic description of the structure and a description of the parameters of materials used in both numerical and experimental modeling. Section 3 presents a description of the theoretical approach to the study of composite elastic, electroelastic and acoustic media in FE modeling. Section 4 covers the numerical simulation of the generator. The results of modal analysis and some output data of PEG parameters as a result of harmonic modeling are presented. Section 5 outlines a description of an experimental setup for testing the operation of PEG under a certain loading. A description of the results of testing PEG under dynamic loading in a certain frequency range is presented. Finally, Section 6 provides the conclusion.

## 2. Description of Model Parameters and Electric Scheme

An axial piezoelectric transducer for converting mechanical energy into electrical energy contains a base plate (7) in the form of a rigid beam structure made of elastic material, on which piezoelectric bimorph (2, 3) elements are glued (see Figure 1). One end of the beam structure is bolted (9) to the base (4) by Fixing supports (5). To the other end, at the base, piezoelectric elements of the cylindrical type (1) are fixed coaxially with the base bar through an L-shaped bar (6). An additional proof mass M (8) is located between the edge of the base, fixed with a bolt (9) and the piezoelectric elements (3).

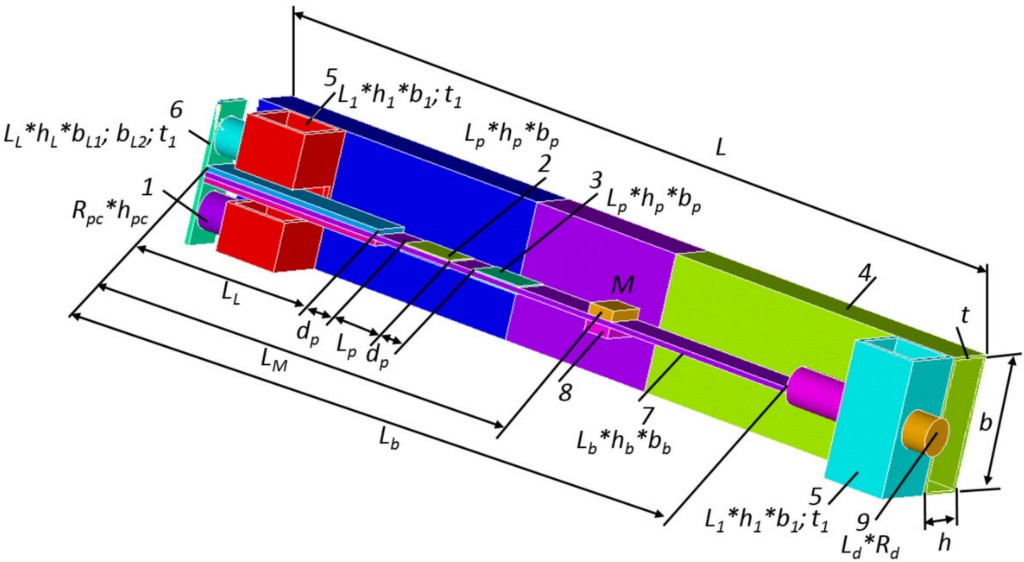

**Figure 1.** Structure scheme of PEG with proof mass: 1—piezocylinder; 2,3—plate piezoelectric elements; 4—rigid base of the generator; 5—PEG fixing supports; 6—L-clamping bar; 7—base; 8—proof mass; 9—screw.

Thin symmetric piezoelectric elements (PE) are polarized in thickness. They are glued to the base console and are arranged in a row (see Figure 2). Characteristics of the dimensions of the PEG elements are presented in the Table 1, the properties of the elements are presented in the Tables 2–4 show the mechanical properties of PE materials. In the full-scale model, we used PE made of CTS-19 material produced at Piezopribor LLC of the Southern Federal University (Rostov-on-Don, Russia). In the simulation, the parameters of materials with equivalent properties presented in [39] are used.

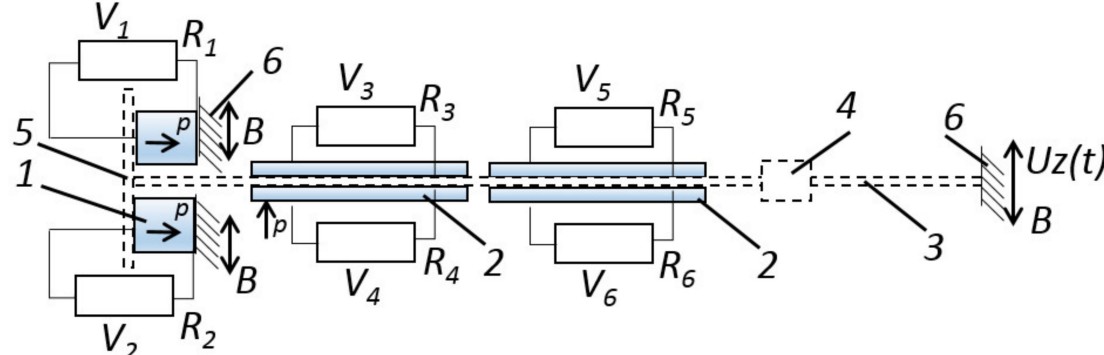

**Figure 2.** Electric scheme of axial-type PEG under active electric load and structure scheme of PEG with proof mass: 1—piezoelectric cylindrical element; 2—piezoelectric bimorph element; 3—substrate; 4—proof mass; 5—clamping bar of PEs (1) fixing; 6—place of PEG fixing. (B) is the movable base; p is the PE's polarization direction.

**Table 1.** Geometric characteristics of the dimensions of PEG elements.

| No. | Name | Geometric Parameters, mm | | | | | |
|---|---|---|---|---|---|---|---|
| 1 | Piezocylinder | $R_{pc}$ | 14 | $h_{pc}$ | 12.5 | | |
| 2, 3 | Piezoelements | $L_p$ | 20 | $b_p$ | 15 | $h_p$ | 0.5 |
| 4 | Rigid base | L | 300 | B | 50 | H | 25 |
| | | t | 2 | | | | |
| 5 | Fixing supports | $L_1$ | 50 | $b_1$ | 25 | $h_1$ | 25 |
| | | t | 2 | | | | |
| 6 | L-clamping bar | $L_L$ | 74 | $b_{L1}$ | 25 | $b_{L2}$ | 24 |
| | | $t_1$ | 15 | $h_L$ | 2 | | |
| | | $d_p$ | 10 | | | | |
| 7 | Base | $L_b$ | 260 | $b_m$ | 14 | $h_m$ | 1.5 |
| 8 | Mass | $L_m$ | 16 | $b_m$ | 10 | $h_m$ | 4 |
| 9 | Screw | $L_d$ | 44 | $R_d$ | 13 | | |

**Table 2.** Mechanical properties of the materials.

| № | PEG Element | Material | $\rho$, kg/m³ | $E \times 10^{11}$, Pa | $\nu$ |
|---|---|---|---|---|---|
| 1 | Piezocylinders | CTS-19 | 7280 | - | 0.33 |
| 2, 3 | Piezoelements | CTS-19 | 7280 | - | 0.33 |
| 4 | Rigid base | duralumin | 2600 | 0.7 | 0.33 |
| 5 | Fixing supports | duralumin | 2600 | 0.7 | 0.33 |
| 6 | L-clamping bar | steel | 7700 | 2.1 | 0.33 |
| 7 | Base | duralumin | 2600 | 0.7 | 0.33 |
| 8.1 | Proof mass | plastic | 1600 | 0,06 | 0.33 |
| 8.2 | Proof mass | duralumin | 2600 | 0.7 | 0.33 |
| 8.3 | Proof mass | steel | 7700 | 2.1 | 0.33 |
| 9 | Screw | steel | 7700 | 2.1 | 0.33 |
| R | Active electric load | resistor | $1\,k\Omega - 2\,M\Omega$ | | |
| | damping value | | $\xi = 0.031$ | | |

**Table 3.** Elastic moduli $C^E_{pq}$ ($\times 10^{10}$ Pa), piezoelectric coefficients $e_{kl}$ (C/m²) and relative permittivity $\varepsilon^\xi_{kk/\varepsilon 0}$ of piezoceramics (based on measurements at room temperature).

| Piezoelement Type | $C^E_{11}$ | $C^E_{12}$ | $C^E_{13}$ | $C^E_{33}$ | $C^E_{44}$ | $e_{31}$ | $e_{33}$ | $e_{15}$ | $\frac{\varepsilon^\zeta_{11}}{\varepsilon_0}$ | $\frac{\varepsilon^\zeta_{33}}{\varepsilon_0}$ |
|---|---|---|---|---|---|---|---|---|---|---|
| CTS-19 | 10.9 | 6.1 | 5.4 | 9.3 | 2.4 | −4.9 | 14.9 | 10.6 | 820 | 840 |

**Table 4.** Elastic compliance of $S^E_{pq}$ ($\times 10^{-12}$ PA), piezoelectric $d_{fp}$ moduli (pC/N) and relative permittivity $\varepsilon^\sigma_{kk}/\varepsilon_0$ of piezoceramics (based on measurements at room temperature).

| Pizoelement Type | $S^E_{11}$ | $S^E_{12}$ | $S^E_{13}$ | $S^E_{33}$ | $S^E_{44}$ | $d_{31}$ | $d_{33}$ | $d_{15}$ | $\frac{\varepsilon^\sigma_{11}}{\varepsilon_0}$ | $\frac{\varepsilon^\sigma_{33}}{\varepsilon_0}$ |
|---|---|---|---|---|---|---|---|---|---|---|
| CTS-19 | 15.1 | −5.76 | −5.41 | 17.0 | 41.7 | −126 | 307 | 442 | 1350 | 1500 |

*Working Principle of the Model*

The proposed model works as follows: When the rigid base (8–10) is exposed to external mechanical forces such as shocks and vibrations, the vibrations are transmitted to the base plate (7), affecting the piezoelectric elements, in which alternating deformations of compression (in cylinders) and tension–compression (in plates) due to the reaction of the supports occur.

Due to the direct piezoelectric effect on the electrodes of additional piezoelectric elements, AC voltage is generated and, therefore, additional electrical energy. The combined use of such elements allows you to increase the output power and conversion efficiency of the converter efficiency. This AC voltage and additional electrical energy can be converted using bridge rectifiers to DC voltage, which is stored in batteries using harvesting energy systems. Piezoelectric elements can be connected in parallel, in series or separately. The choice between the types of connections of elements depends on the device that needs to be powered: if a higher output voltage is required, then a serial connection should be selected, and if a higher output current is required, then a parallel connection.

A schematic electrical diagram of a PEG connection with a resistive load is shown in Figure 2. The resistive load is supplied to each PE individually. Voltage is found at the contact points of the resistor. In numerical simulation, voltage is calculated as the difference of its amplitudes at the nodes of the FE element presented, with an option indicating the type of resistor. The added weight can vary from 3 to 25 g. In the experiment, we used the masses M = 3.71; 6.13 g. The proof mass was located at a distance of Lm = 150 to 230 mm to the left edge of the generator (Figure 1).

## 3. Theoretical Description of the Model of Composite Elastic, Electroelastic and Acoustic Media by FE Simulation

The energy storage PEG is a composite elastic and electroelastic body. It is assumed that the device performs elastic small oscillations in a moving coordinate system. The rectilinear vertical motion of this system in the area of fixation is given by the law for steady oscillations:

$$u(t) = \overline{u}\, e^{i\omega t} \tag{1}$$

Under these conditions, a sufficiently adequate mathematical model of the operation of the device is the initial-boundary value problem of the linear theory of electroelasticity [40,41].

In the general formulation, the equations for a piezoelectric medium are written as:

$$
\begin{aligned}
\rho \ddot{u}_i + \alpha \rho \dot{u}_i - \sigma_{ijj} &= f_i; \\
D_{i,i} &= 0; \\
\sigma_{ij} &= c_{ijkl}\left(\varepsilon_{kl} + \beta \dot{\varepsilon}_{kl}\right) - e_{ijk} E_k; \\
D_i + \varsigma_d \dot{D}_i &= e_{ikl}\left(\varepsilon_{kl} + \varsigma_d \dot{\varepsilon}_{kl}\right) - \epsilon_{ik} E_k; \\
\varepsilon_{kl} &= (u_{kl} + u_{lk})/2; \\
E_k &= -\varphi_k;
\end{aligned}
\tag{2}
$$

where ρ is the material density;

$u_i$ are the components of the displacement vector;

$\sigma_{ij}$ are the components of the stress tensor;

*f* are the components of the vector of the density of mass forces;

$D_i$ are the components of the electric induction vector;

$c_{ijkl}$ are the components of the fourth rank tensor of the elastic moduli;

$e_{ikl}$ are the components of the third rank tensor of piezoelectric coefficients;

$\varepsilon_{kl}$ are the components of strain tensor;

$E_k$ are the components of the electric field vector;

$\varphi_k$ is the electric potential;

$\epsilon_{ik}$ are the components of the dielectric constants tensor;

$\alpha$, $\beta$, $\zeta$ are non-negative damping coefficients (the value of $\zeta_d$ is used in ANSYS software).

For elastic medium, we have:

$$
\begin{aligned}
\rho \ddot{u}_i + \alpha \rho \dot{u}_i - \sigma_{ijj} &= f_i; \\
\sigma_{ij} &= c_{ijkl}\left(\varepsilon_{kl} + \beta \dot{\varepsilon}_{kl}\right); \\
\varepsilon_{kl} &= (u_{kl} + u_{lk})/2.
\end{aligned}
\tag{3}
$$

Since harmonic analysis is used in the calculations, the following actions are performed for the corresponding components of the equations:

$$
\begin{aligned}
\dot{u} &= i\omega u; \\
\ddot{u} &= i\omega u^2.
\end{aligned}
\tag{4}
$$

To solve the problem, the following mechanical and electrical boundary conditions are accepted.

Boundary conditions are given in the form of a displacement field on the boundary $S_u$

$$
u_i|_{S_u} = u_i^0
\tag{5}
$$

Boundary conditions in the form of a vector of surface stresses $p$

$$
t = \sigma n|_{S_t} = P
\tag{6}
$$

The boundary conditions on the electrodes of the piezoelectric element $S_E = \bigcup_k S_{E_k}$ are given as

$$
\varphi|_{S_{E_k}} = \varphi_{0k}
\tag{7}
$$

Boundary conditions on non-electrode sections S_D, at the intersection of the corresponding areas $S = S_E \bigcup S_D$

$$
D_n|_{S_D} = 0
\tag{8}
$$

The attenuation coefficient parameters are between the frequencies и$f_{r1}$ and и$f_{r2}$. It is assumed that within the framework of the experiment, the change in the damping parameters $\alpha$ and $\beta$ will be minimally changeable. In the FE package ANSYS, the damping parameters were described in the form

$$
\alpha = \frac{2\pi f_{r1} f_{r2}}{Q(f_{r1} + f_{r2})}, \quad \beta = \frac{1}{2\pi Q(f_{r1} + f_{r2})}
\tag{9}
$$

From the experiment, the quality factor Q is found from the expression

$$
Q = \frac{\omega}{\Delta \omega}
\tag{10}
$$

where $\Delta \omega$ is the width of the resonance curve. This is found at the corresponding resonance $\omega$.

The solution of the set system of Equations (1)–(8) is solved, taking into account the initial boundary conditions for non-stationary problems [40]. The elements of solving the system of Equations (1)–(8) within the framework of modeling can be implemented in the Ansys complex. Within the framework of the presented studies, a number of direct calculations were carried out in the form of a modal analysis with obtaining natural modes of oscillations and frequency characteristics. When carrying out harmonic analysis, the

electric potential on the electrodes was calculated relative to the zero potential on certain surfaces. The displacement of the base by 0.01 mm was taken as the perturbation parameter.

## 4. Modeling

Within the framework of the presented studies, modal and harmonic analyses of PEG oscillations were carried out during modeling. When modeling, the value of the attached mass M= {0.05; 1; 3; 5; 7; 9} gr, as well as its position Lm = {150; 170; 190; 210; 230} mm on a flexible base plate is used. The value of the active load was taken as R = 1000 $\Omega$. At the stage of harmonic analysis, the vibration amplitudes were recorded at a displacement of the PEG rigid fastening zone by {0.001...0.01} mm.

The next task was to consider the harmonic modeling of PEG oscillations with a constant displacement of the generator housing attachment zone by 0.01 mm and variations in material properties (duralumin: $E_1$ = 0.7 $10^{11}$, Pa, $\rho_1$ = 2600 kg/m$^3$, $\nu^1$ = 0.33; and fiberglass: $E_2$ = 0.06 $10^{11}$, Pa, $\rho_2$ = 1600 kg/m$^3$, $\nu_2$ = 0.33) and corresponding thickness options h = {1; 2} mm base plate. In this case, the attached mass was fixed in the position $L_m$ = 150 mm and its value was M = 0.5 gr, 30 gr. As an active load in the form of resistance, R for each piezoelectric element was taken, equal to 1000 $\Omega$.

The natural frequencies, the oscillation amplitude of the center of the flexible base plate of the generator and the output voltage taken from the piezoelectric elements were considered as output parameters.

The simulation was carried out in FE software ANSYS. Figure 3 shows the FE model of the PEG. PEG consists of a base in the form of a strip, clamped on one of the sides. During the modeling, elements of the SOLID92 type with a tetraidal structure were used, which facilitated the partition of the model. When modeling the thin walls of the planks, it was assumed not to consider deformations over the thickness of the structure. As a result, the value of the smallest edge of the FE in the form of a tetrahedron was taken, equal to the wall thickness of the structural element. Modeling of piezoelectric elements was carried out using FE type SOLID5. Modeling of elements in the form of active resistance was carried out using the FE CIRCU94 with the resistor option. The PE of a three-dimensional structure in the form of piezocylinders (PE PC) was divided by a FE mesh with an edge size multiple of 0.15 of the piezocylinder height. In a PE with a planar structure (PE PS), the size of the partitioning of the FE mesh by thickness was a multiple of the value equal to the thickness of the PE. The division was in the form of triangular prisms. The direction of polarity in a cylindrical PE was taken along the main axis of the PE, coaxially with the PEG, in a plate-shaped PE along the thickness. The polarization vector in the bimorph for the upper and lower PEs was directed along the normal to the surface. The number of FE elements when splitting the model was more than 33,000 nodes greater than 62,500.

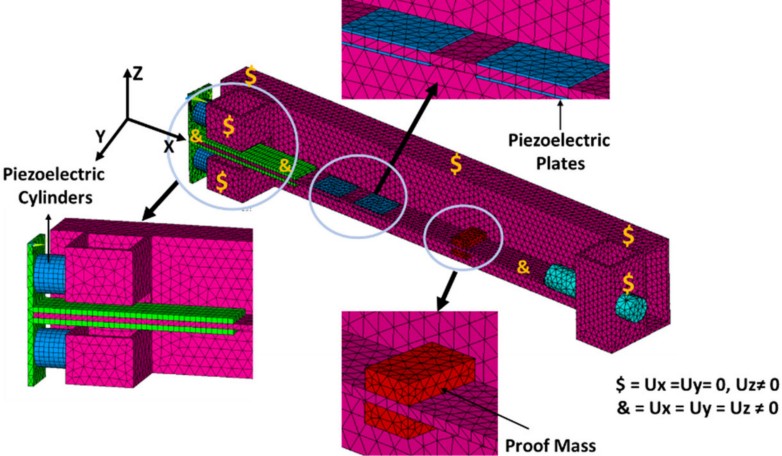

**Figure 3.** FE-model of PEG of axial type and boundary conditions.

At the first stage, modal PEG analysis was carried out. Figure 4 shows the four vibration modes of the model. With this configuration, the model had a first design frequency of 259 Hz. The vibration mode for the first mode had a prevailing bending character for the generator warp bar. Vibration modes 1, 4 and 8 correspond to (1,2,3) flexural vibration modes of the PEG base plank in the vertical direction. The second mode shape was obtained at 525 Hz. It was assumed that bending deformations of the bar in the smallest plane of rigidity (in the vertical direction) would excite the highest output stress in all PEs.

Mode-1, 259 Hz

Mode-2, 525 Hz

Mode-4, 684 Hz

Mode-8, 1849 Hz

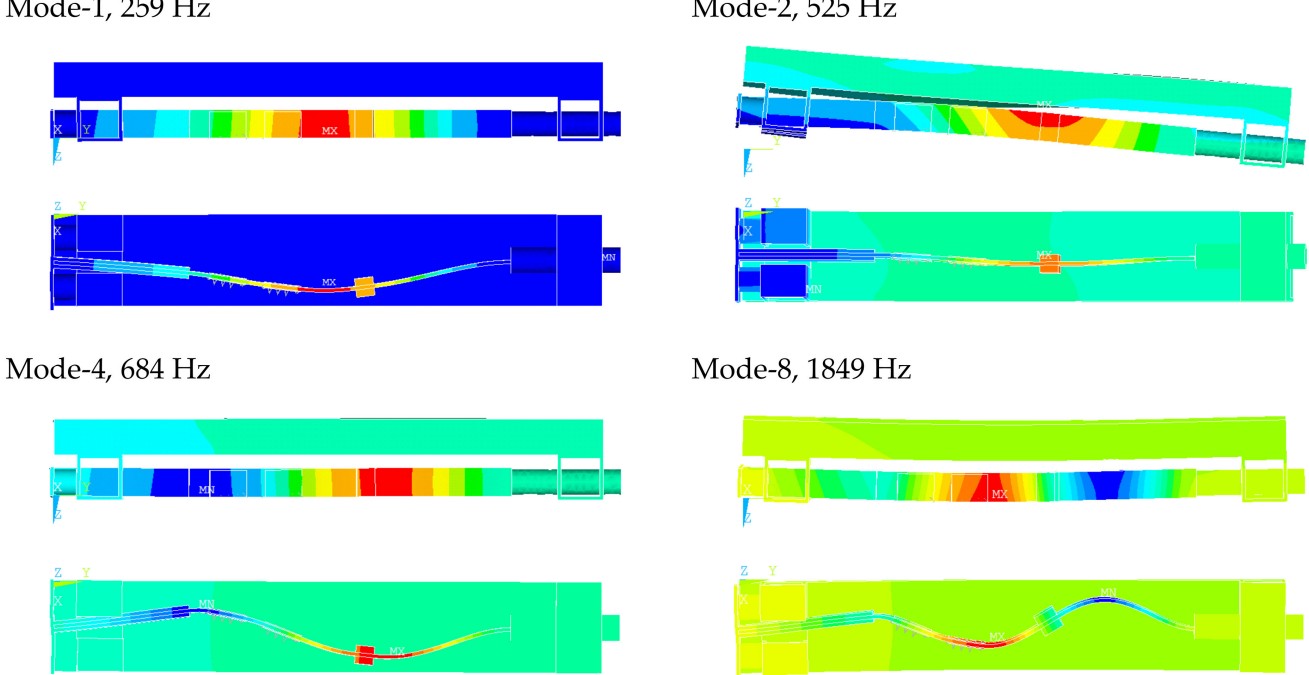

**Figure 4.** Natural frequencies and mode shapes of the PEGs.

Investigations of oscillations of axial-type PEG under harmonic action were carried out. The harmonic action was calculated under the action of a uniformly applied acceleration of 10 m/s$^2$ for all structural units at the corresponding harmonics. For the first and second vibration modes, the dependences of their value on the mass and location of the load were calculated. The corresponding graphs (Figure 5) show that the lowest frequency values are in the range of 212–220 Hz for the first vibration mode and values of 501–510 Hz can be achieved with the central location of the load and its maximum mass of 9 g. This, at the point of attachment of the load, will be maximum, which is shown in Figure 6. Figure 7 shows the calculated parameters of the dependence of the output voltage at the electrodes of piezocylinders ($U_1$) and piezoelectric elements in the form of plates ($U_2$ and $U_3$), respectively, with their calculated arrangement from left to right (see Figure 1). In the calculations, the value of the active load of the corresponding PE was taken to be 1000 Ω for one vibration mode.

A numerical experiment was set up to establish the dependence of the first oscillation frequency on the associated proof mass 8 (Figure 1). The variable parameters were the modulus of elasticity $E_i$ of the base 7 (Figure 1) and its thickness $h_i$. The proof mass varied within M = 0.5 gr, 30 gr. The place of fixation-proof mass $L_m$ = 150 mm is the point in the middle of the base. Accordingly, at fixed dimensions of the load, the specific density of the material was calculated. It was assumed that the most sensitive characteristic for changing the natural frequency of PEG vibrations is the value of the mass-proof mass. The modulus of elasticity of the volume-proof mass was taken equal to the properties of the base. The base modeling was considered, using the following properties of duralumin for calculations: $E_1$ = 0.7 10$^{11}$, Pa, $\rho_1$ = 2600 kg/m$^3$, $\nu_1$ = 0.33; and using the properties

of fiberglass: $E_2 = 0.06 \cdot 10^{11}$, Pa, $\rho_2 = 1600$ kg/m$^3$, $\nu_2 = 0.33$. The thickness of the base (7) was assumed to be $h_1 = 2$ mm and $h_2 = 1$ mm for calculations. Thus, four options for the layout of parameters for modeling were used: 1—$E_1$, $h_1$; 2—$E_1$, $h_2$; 3—$E_2$, $h_1$; 4—$E_2$, $h_2$. The results of numerical calculations obtained on the basis of the modal analysis carried out are presented on Figure 8. An analysis of the obtained frequency dependences shows the following: with an increase in mass for all design simulation options, the first natural frequency decreases. Therefore, with a conditionally small mass of 0.5 gr, the first natural frequency for options was 1—287.4 Hz; 2—154 Hz; 3—122.8 Hz; 55.3 Hz. In this case, the first natural frequency with a mass of 30 gr was, respectively, 1—143.4 Hz; 2—59.7 Hz; 3—50 Hz; 18.5 Hz. In a comparative analysis for all calculation options, the first frequency changed, respectively, for variations, more than 1–2 times, 2–2.57 times, 3–2.45 times and 4–2.97 times. Thus, with various initial parameters of the properties of the base model, the use of this PEG is possible in various loading ranges, both in the low-frequency region up to 50 Hz and in the region of higher frequencies, using only the first oscillation mode up to 287 Hz. The loading mode in the region of more than 50 Hz involves the use of devices for mechanical excitation of oscillations, for example, rotary motors with magnetic media. In addition, these modes of operation of the PEG can be used as the use of the PEG in the form of vibration sensors of the impulse action on the structure in a certain fixed frequency range.

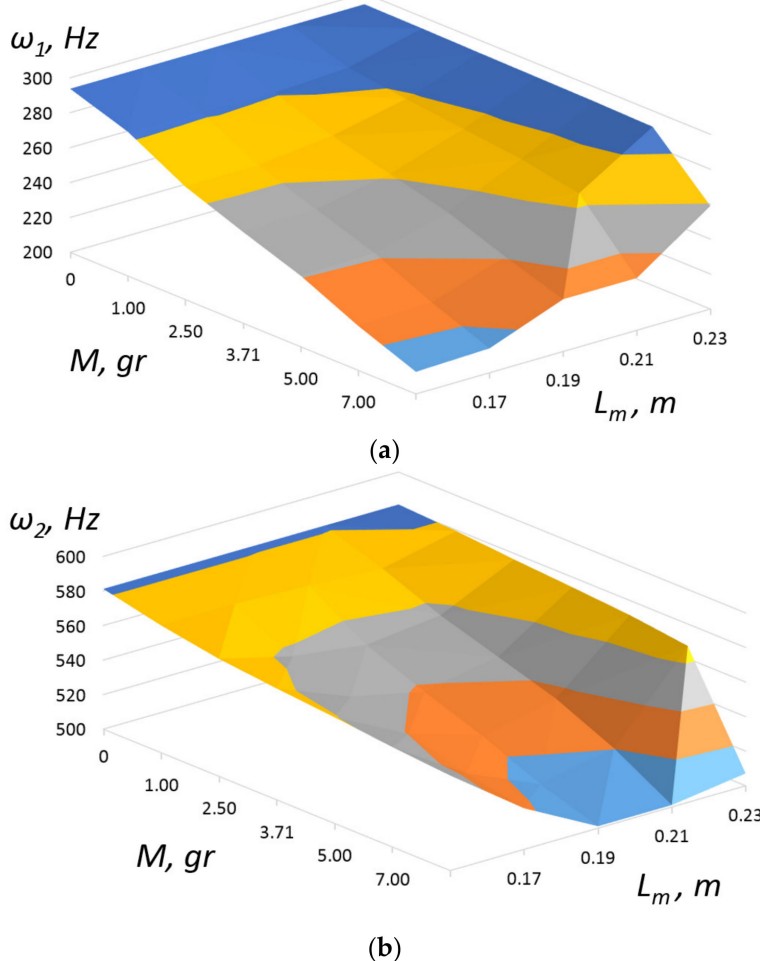

**Figure 5.** Dependence of 1 (**a**) and 2 (**b**) natural frequencies of PEG on the magnitude and location of proof mass at the lowest frequency.

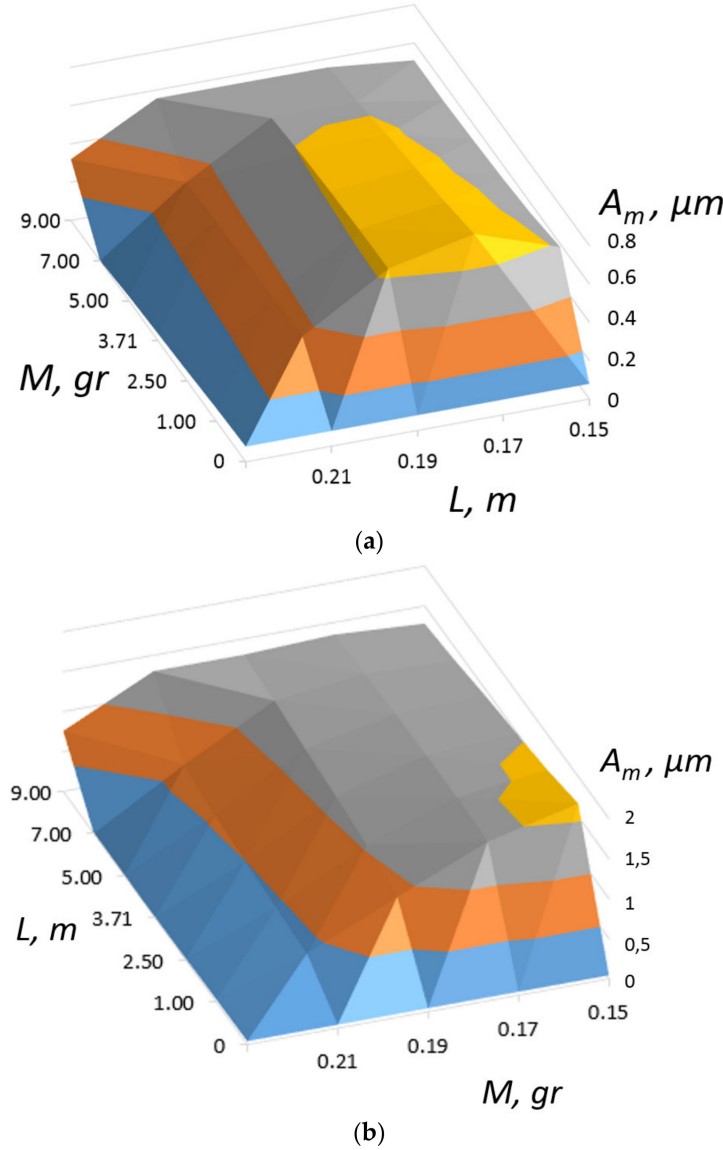

**Figure 6.** Dependence of 1 (**a**) and 2 (**b**) natural frequencies of PEG on the magnitude and location of proof mass.

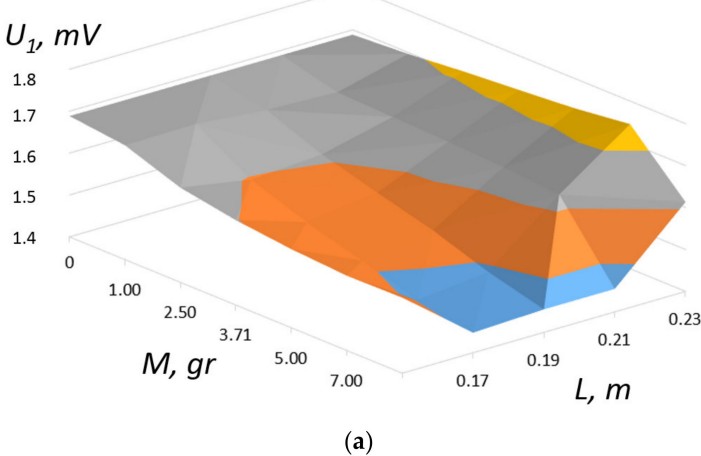

(**a**)

**Figure 7.** *Cont.*

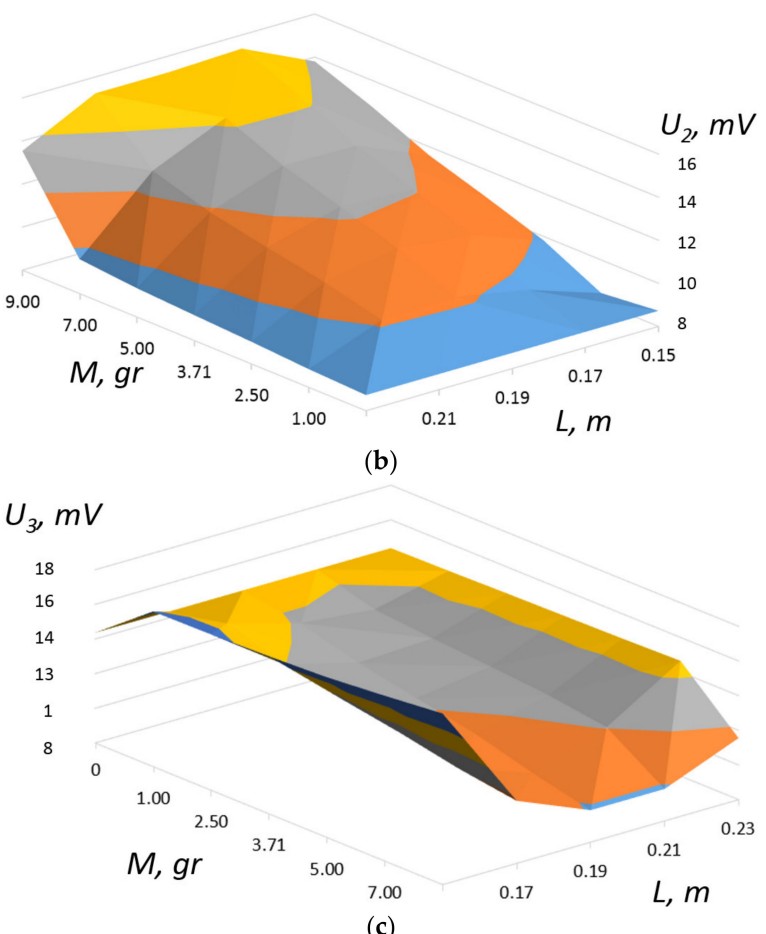

**Figure 7.** Dependence of the output voltage U on the electrodes of the corresponding PE at an active load of 1000 $\Omega$ for 1 vibration mode. Respectively, for (**a**) piezocylinders ($U_1$) and (**b**,**c**) piezoelectric elements in the form of plates ($U_2$ and $U_3$).

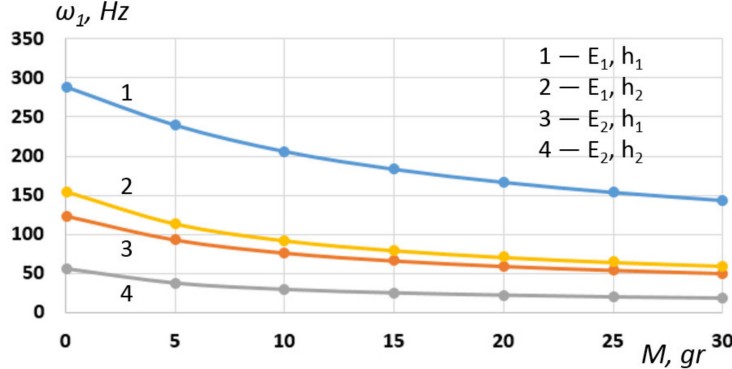

**Figure 8.** Dependence 1 of the first natural frequency of the PEG on various variations in the parameters of the base plank ($E_i$—the modulus of elasticity and $h_i$—the height of the base).

## 5. Experimental Probe

To conduct field studies of the operation of an axial-type PEG and obtain primary results for assessing its output parameters, a laboratory test setup, LTS -01, was created with certified testing devices. A structural diagram of its work was built and a description of the setup and a research methodology was prepared.

The LTS -01 laboratory test setup for studying the PEG output characteristics is shown in Figure 9. The direct piezoelectric effect on the electrodes of additional piezoelectric elements and AC voltage was generated and, therefore, additional electrical energy. The

combined use of such elements allows you to increase the output power and conversion efficiency of the converter efficiency. This AC voltage and additional electrical energy can be converted using bridge rectifiers to DC voltage, which is stored in batteries using harvesting energy systems. Piezoelectric elements can be connected in parallel, in series or separately. The LTS -01 laboratory test setup for studying the output characteristics of an axial-type PEG consists of an exciter of mechanical vibrations—an electromagnetic vibrator VEB Robotron 11,077 (4), on a work Table 5, on which the studied PEG sample was installed. The sample under study is a plate (14) fixed on the base (12). On the plate (12), there are two pairs of piezoelectric elements (17) made in the form of a bimorph. At the left end of the plate base, there were cylindrical-type piezoelectric elements having a longitudinal arrangement along the main PEG axis. The other end of the generator base was rigidly bolted, thereby clamping and effectively fixing the opposed generator base with cylindrical piezoelectric elements.

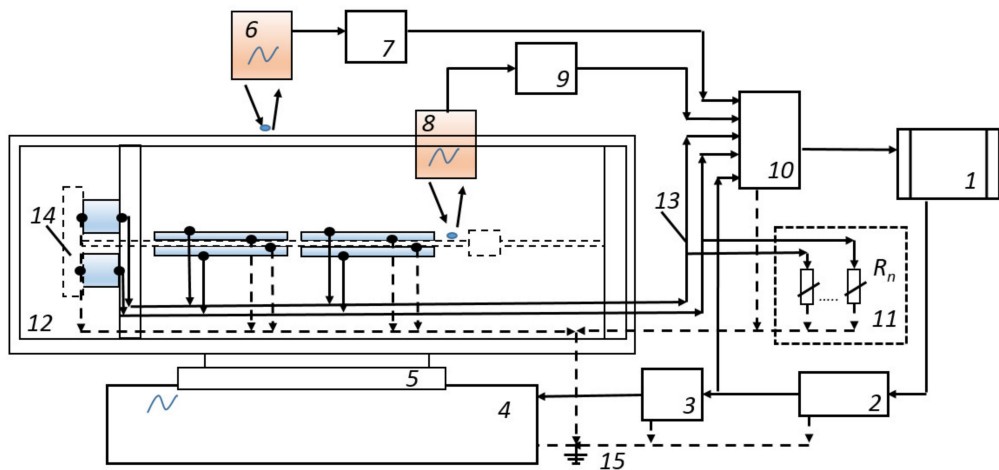

**Figure 9.** Block scheme of the LTS -01 laboratory test setup for studying the PEG output characteristics: 1—computer with monitor; 2—AFG 3022B Tektronix signal generator; 3—power amplifier LV-102; 4—electromagnetic vibrator VEB Robotron 11077; 5—vibrator working table for fixing the model; 6—RF603 optical linear displacement transducer; 7—optical sensor switching unit (6); 8—optical linear displacement transducer optoNCDT; 9—optical sensor switching unit (8); 10—external ADC/DAC E14-440D module; 11—bank of electric load resistances Rl; 12—the base of the investigated object; 13—matching path; 14—piezoelectric generator of axial type; 15—grounding path.

**Table 5.** Result comparison.

| Mode | Stimulation | Experimental | Error % |
|---|---|---|---|
| 1 Natural Frequency (NF) | 283 Hz | 302 Hz | 6.29 |
| 2 Natural Frequency (NF) | 581 Hz | 587 Hz | 1.1 |
| Deflection at 1 NF | 1.59 mm | 1.5 mm | 5.6 |
| Deflection at 2 NF | 0.72 mm | 0.7 mm | 2.7 |

There was a proof mass on the base bar (14), with the help of which it was possible to easily adjust the frequency characteristics of the generator in certain ranges. With the help of optical sensors of mechanical displacements (6) and (8) and their matching devices (7) and (9), data on the vibration amplitudes of the corresponding PEG points were transmitted to the computer. The optical sensor (6) of the REF603 type was located above the point in the center of the rigid base of the PEG and transmitted information about the vibrations of the working table plate of the VEB Robotron 11,077 electromagnetic vibrator (4). The optoNCDT optical linear displacement transducer (7) was located above the PEG base plate and could transmit a voltage linearly, proportional to the oscillation amplitudes at

predetermined fixed points of the base, through a conductive loop. Within the framework of the experiment, the vibrations of the central region of the base plate were recorded.

The measuring part of the LTS -01 consisted of an analog-to-digital converter—an external ADC/DAC E14-440D module from L-Card (10) and a personal computer (PC) (1). This external module was used to record the voltage at the electrical contacts of the piezoelectric elements under an active electrical load Rl (11), as well as the voltage at the electrical contacts of the laser displacement sensors (6) and (8) and the AFG 3022B Tektronix master signal generator (2).

### 5.1. The Principle of Operation of the Laboratory Test Setup LTS -01

The process of measuring the frequency response of the oscillatory process of the PEG model and recording its output characteristics was as follows and is shown in Figure 10. On a personal computer (1), two programs were launched: (i) a program for recording voltage parameters on an E14-440D ADC/DAC module from L-Card (10) PowerGraph and (ii) a swept signal generation program with its own developed software [42]. The signal had a fixed output oscillation amplitude with a voltage of 2 V. The process of sweeping (enumerating) the signal was carried out by alternating the excited frequencies from 1 Hz to 1000 Hz. Excitation of each vibration frequency was carried out for 0.5 sec with a pause of 0.5 sec between the transition to another frequency, with a step between frequencies of 1 Hz. The signal could be transmitted via the audio path or USB channel to the AFG 3022B Tektronix signal generator (2), thereby being recorded or generated on it. The PowerGraph program allows high-quality recording and unloading of the measured signal voltage to the ADC module (10). The generated signal was fed through the current-carrying path to the electromagnetic forced oscillation exciter. Voltage was supplied from the oscillator of the sweeping frequency of the device with a stable amplitude and a linearly varying frequency.

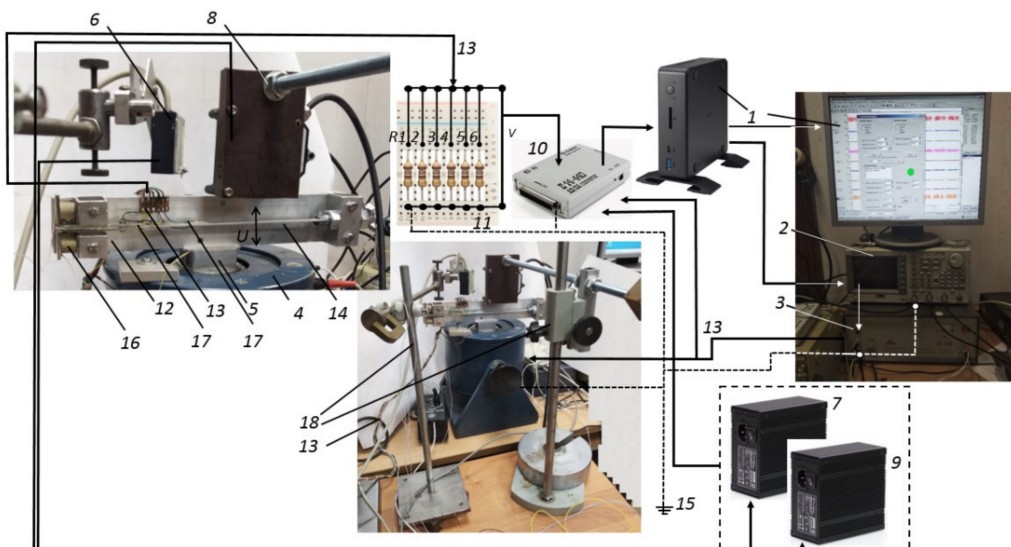

**Figure 10.** General view of laboratory test setup LTS -01 for studying the output parameters of PEG. 1—computer with monitor; 2—AFG 3022B Tektronix signal generator; 3—power amplifier LV-102; 4—electromagnetic vibrator VEB Robotron 11077; 5—vibrator working table for fixing the model; 6—RF603 optical linear displacement transducer; 7—optical sensor switching unit (6); 8—optical linear displacement transducer optoNCDT; 9—optical sensor switching unit (8); 10—external ADC/DAC E14-440D module; 11—bank of the active electric load resistance R1–R6 of the corresponding PEG elements; 12—the base of the investigated object; 13—matching path; 14—piezoelectric generator of axial type; 15—grounding path; 16—piezoelectric cylindrical type; 17—piezoelectric elements of the plate type; 18—tripods for mounting optical sensors.

The AC voltage value was amplified by the power amplifier (3) and supplied to the vibrator (4); the resulting output voltage from the piezoelectric generator was loaded with AC electrical resistance and supplied to one of the ADC channels (10) of the external module. The values of these voltages were reproduced on the computer monitor screen (1) in the form of an amplitude–time characteristic (ATC). On the working Table 5 of the electromagnetic forced oscillation exciter VEB Robotron 11,077 (4), the investigated PEG sample of the axial type was fixed with a rigidly bolted connection. When voltage was applied to the vibrator (4), predetermined amplitude displacements occured in the vertical direction of the working Table 5. The vibration amplitude in a certain frequency range was linearly proportional to the vibration excitation frequency. The resulting vertical displacements of the working Table 5 set the PEG in motion. Mechanical vibrations of various parts of the generator generated voltage across all piezoelectric elements. The piezoelectric elements were connected to the grounding path (15) and in parallel with the active electric load resistance bank (11) of the corresponding rating. There were six piezoelements and, accordingly, each of them had its own individual electrical path. The voltage excited on the piezoelectric elements was recorded using the ADC module (10). Using the ADC module (10), the voltage values from the optical displacement sensors (6) and (8) were also recorded. Subsequent signal processing was performed by software using an external ADC/DAC E14-440 module from L-Card, a personal computer and PowerGraph software. Observation of the shape of the exciting signal and signals from the sensors was performed on the monitor screen and can be duplicated using a digital oscilloscope. Accurate measurement of the signal frequency was performed using the Fourier analysis module in the software. Adjustment and calibration of receiving sensors were carried out using a measuring microscope. Test results are stored in the PC's memory in digital and text format; signal elements can be duplicated in various text editor modules.

Main technical characteristics of the LTS -01 test setup:

(i)      a range of measurable lateral displacements of the PEG substrate from 0 to 5 mm;
(ii)     frequency of forced oscillations from 1 to 1000 Hz;
(iii)    linear range of forced vibration amplitudes from 20 to 1000 Hz;
(iv)    the sensitivity limit of the optical displacement sensor is not less than 5 µm;
(v)     electric voltage at the input of the electromagnetic exciter of oscillations from 0.1 to 10 V.

### 5.2. PEG Results Validation

The numerical simulations of harmonic analysis of the PEGs were carried for various configurations. The experimental results were taken care of for PEGs. The simulation and experimental results are shown in Figure 11 and summarized in Table 5. The simulation and experimental results were carried in case the proof mass had been located at 230 mm. The analysis of the comparison of the oscillation parameters of harmonic analysis showed a satisfactory coincidence of experimental data on oscillations and the results of numerical studies of their own frequencies and corresponding oscillation amplitudes for the first two modes within 6%.

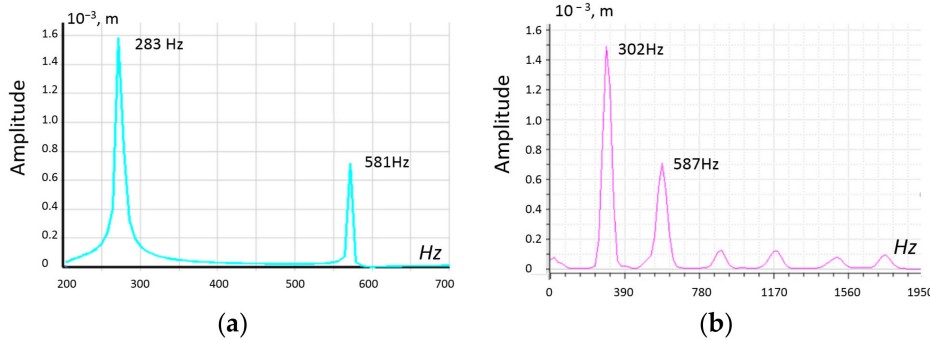

**Figure 11.** Result comparison, (**a**) numerical simulation and (**b**) experimental simulation.

### 5.3. Method of Natural Modeling of the Oscillatory Processes of PEG

The sequence for determining the parameters of PEG characteristics was as follows. At the first stage, a preliminary assessment of the difference in natural frequencies obtained in the experiment and simulation was carried out. By formula (11), the discrepancy of natural frequencies was calculated:

$$\Delta = [(\omega_{\text{Exp}} - \omega_{\text{Sol}})/\omega_{\text{Sol}}] \times 100\%. \tag{11}$$

Table 6 shows a comparison of the first and second natural harmonics of oscillations, with an experimental value of the mass of the load of 3.71 g at its various locations. The active load for each PE was taken, both in the experiment and in the calculation of 10 kΩ. The analysis of the obtained results showed a satisfactory coincidence of the values of the natural frequencies (the difference was no more than 5%).

**Table 6.** Comparison of the experimental and numerical values 1 and 2 of the natural frequencies of PEG oscillations at an active load of PE of 10 kΩ and a load value of 3.71 g at different positions.

| | Location of Proof Mass $L_m$, mm | | | | |
|---|---|---|---|---|---|
| | 150 | 170 | 190 | 210 | 230 |
| | 1 natural frequency, Hz | | | | |
| Experiment $\omega_{\text{Exp}}$ | 255.0 | 258.0 | 267.0 | 288.0 | 302.0 |
| Calculation $\omega_{\text{Sol}}$ | 252.2 | 253.0 | 261.2 | 274.8 | 292.2 |
| $\Delta$, % | 1.1 | 2.0 | 2.2 | 4.8 | 4.0 |
| | 2 natural frequency, Hz | | | | |
| Experiment $\omega_{\text{Exp}}$ | 581.0 | 576.0 | 560.0 | 563.0 | 587.0 |
| Calculation $\omega_{\text{Sol}}$ | 564.4 | 547.5 | 538.1 | 546.0 | 574.2 |
| $\Delta$, % | 2.9 | 5.2 | 4.1 | 3.1 | 2.2 |

First, the frequency response of the piezoelectric generator was recorded, then the dependence of the output voltage and power of the PEG at the resonant frequency on the value of the active electrical load and proof mass on the value of the output voltage of the PEG and frequency characteristics was investigated. In the test PEG sample, fixed on a vibrating plate, transverse bending vibrations were sequentially excited in the frequency range from 0 to 1000 Hz (the generator voltage amplitude did not change in this case) value of electrical load resistance Rl from 10 kΩ to 2 MΩ. The output power was calculated for the obtained voltage characteristics for 1, 2 and 3 PE.

Figure 12 shows examples of visualization of the process of measuring the output characteristics of a PEG during sweeping. This analysis reveals the parameters of the output voltage on the PE electrodes at different loads. During the sweep, two points were chosen to obtain the estimated parameters of the PEG operation: (i) 39 Hz and (ii) 107 Hz. At a frequency of 107 Hz, the oscillation frequencies of the system elements were found, causing a resonant increase in oscillations. For a frequency of 39 Hz, the oscillation amplitude of the rigid base of the generator (4) was 0.266 mm, while the oscillation amplitude of the base (7) center was 1.683 mm. For a frequency of 107 Hz, the oscillation amplitude of the rigid base of the generator (4) was 0.035 mm, while the oscillation amplitude of the base (7) center was 0.356 mm. The results are shown in Tables 7 and 8. Data analysis shows that the maximum output voltage of 1.31 V for cylindrical PE occurs at a frequency of 39 Hz and a load of 2 MΩ. At the same time, for PE in the form of plates, the highest voltage is achieved at an active load of 150 kΩ and a frequency of 39 Hz. Figures 13 and 14 show the graphs of the dependence of the output voltage and peak output power for all types of PE used. Analysis of the output power shows that with a large value of resistance, it drops. In this case, the maximum output power can be obtained at a load of up to 51 kΩ.

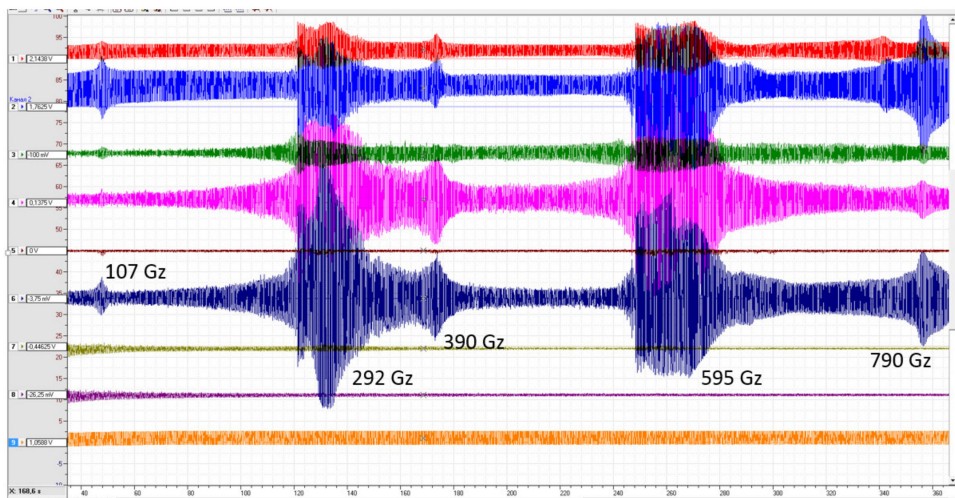

**Figure 12.** Visualization of the process of measuring the PEG output characteristics during sweeping.

**Table 7.** Experimental measurements of the voltage amplitude on the PE electrodes at the corresponding vibration frequencies and the corresponding active electric load.

| PE N | Voltage Amplitude (V) across the Electrodes at Oscillation Frequency of 39 Hz and the Corresponding Active Electric Load R (kΩ) | | | | | |
|---|---|---|---|---|---|---|
| | 10 kΩ | 51 kΩ | 75 kΩ | 150 kΩ | 300 kΩ | 2000 kΩ |
| 1 | 0.14625 | 0.294872 | 0.355079 | 0.392063 | 0.560649 | 1.317638 |
| 2 | 0.06685 | 0.096381 | 0.100432 | 0.222399 | 0.207643 | 0.187802 |
| 3 | 0.06505 | 0.085808 | 0.077982 | 0.1905 | 0.183093 | 0.180334 |
| | Voltage Amplitude (V) across the Electrodes at an Oscillation Frequency of 107 Hz and the Corresponding Active Electric Load R (kΩ) | | | | | |
| | 10 kΩ | 51 kΩ | 75 kΩ | 150 kΩ | 300 kΩ | 2000 kΩ |
| 1 | 0.043576 | 0.094412 | 0.061982 | 0.117833 | 0.127969 | 0.467393 |
| 2 | 0.032606 | 0.043102 | 0.06355 | 0.136766 | 0.111208 | 0.081129 |
| 3 | 0.02439 | 0.082987 | 0.078731 | 0.109417 | 0.106021 | 0.106581 |

**Table 8.** Experimental measurements of the output power peak on the PE electrodes at the corresponding vibration frequencies and the corresponding active load.

| PE N | Peak Power (µW) at the PE Electrodes at an Oscillation Frequency of 39 Hz and the Corresponding Active Load R (kΩ) | | | | | |
|---|---|---|---|---|---|---|
| | 10 kΩ | 51 kΩ | 75 kΩ | 150 kΩ | 300 kΩ | 2000 kΩ |
| 1 | 2138.9 | 1704.9 | 1681.1 | 1024.8 | 1047.8 | 868.1 |
| 2 | 446.9 | 182.1 | 134.5 | 329.7 | 143.7 | 17.6 |
| 3 | 423.2 | 144.4 | 81.1 | 241.9 | 111.7 | 16.3 |
| | Peak power (µW) at the PE Electrodes at an Oscillation Frequency of 107 Hz and the Corresponding Active Load R (kΩ) | | | | | |
| | 10 kΩ | 51 kΩ | 75 kΩ | 150 kΩ | 300 kΩ | 2000 kΩ |
| 1 | 189.89 | 174.78 | 51.22 | 92.56 | 54.59 | 109.23 |
| 2 | 106.31 | 36,43 | 53.85 | 124.7 | 41.22 | 3.29 |
| 3 | 59.49 | 135.04 | 82.65 | 79.81 | 37.47 | 5.68 |

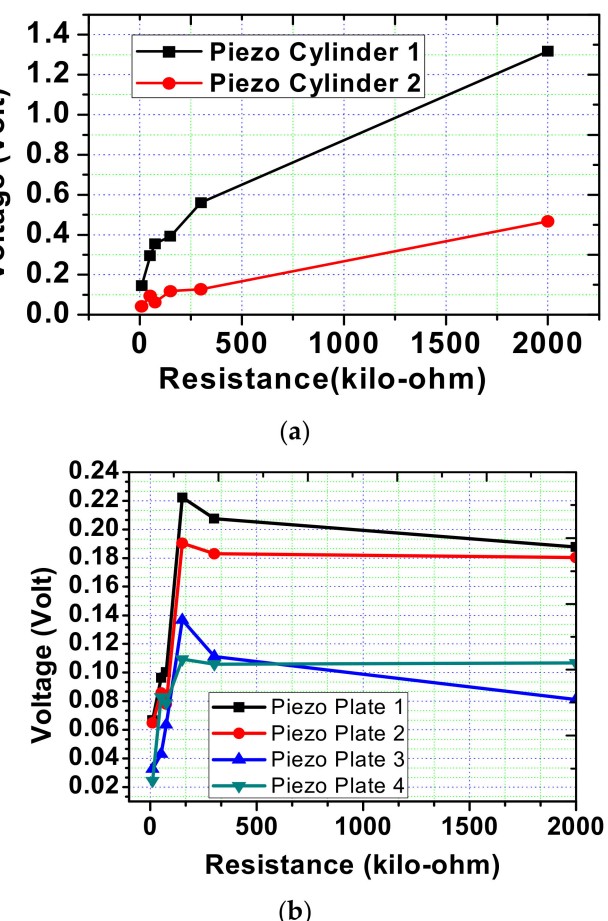

**Figure 13.** Dependence of the output voltage on the magnitude of the load. (**a**) Piezocylinders and (**b**) piezoplates.

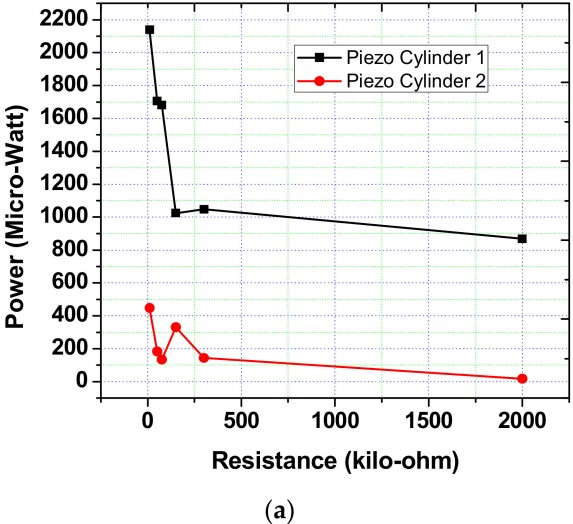

(**a**)

**Figure 14.** *Cont.*

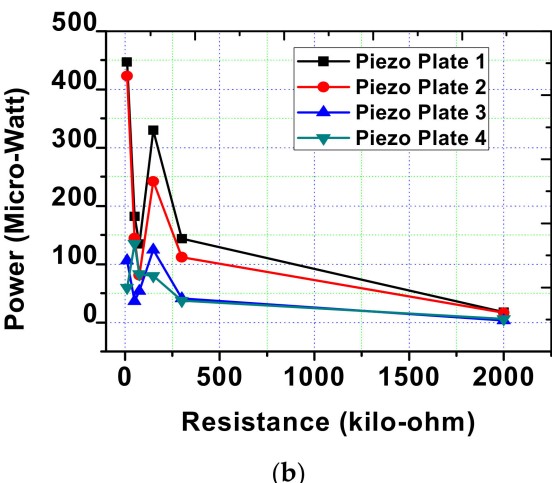

(**b**)

**Figure 14.** Dependence of the output power P on the value of the electric load, (**a**) Piezocylinders and (**b**) piezoplates.

## 6. Conclusions

As a result of the research carried out, a finite element, by using ANSYS software and experimental approaches to modeling oscillations of a new type of piezoelectric generator (PEG) with proof mass, and an active base were developed; as for the latter, cylindrical piezoelements located along the axis of the generator were used. The results of modal and harmonic analysis of oscillations are presented. In this case, the generation of energy in the cylindrical-type piezoelectric elements occurred due to the transfer of compression forces to the piezoelectric element at the base of the PEG upon excitation of structure vibrations. The piezoelectric elements of the plate type, made in the form of a bimorph, used the potential energy of bending vibrations of the PEG on the PEG base bar. The presented technique for the experimental analysis of vibrations, as well as the laboratory setup developed, made it possible to obtain experimental results of the output characteristics of the piezoelectric generator under low-frequency loading, which differ from the finite element results within 5%.

The presented PEG has frequencies of the first vibration mode in the range from 210 Hz to 300 Hz with the corresponding proof mass position. The scope of this generator can be attributed not only to high-frequency loading ranges. On the experimental setup, created within the framework of the research, a variant of low-frequency loading at frequencies of 39 and 107 Hz was considered, at which the highest output voltage for frequency 39 Hz for a cylindrical PE was 0.14 V and the output power was $P_1$ 2138.9 µW. For the PE plate type for this frequency 39 Hz, the maximum peak power was $P_2$ 446.9 µW and $P_3$ 423.2 µW. When conducting a comparative analysis with literature data, the analysis shows that the output parameters of the generator are at an average level and may require further modification of the design.

The analysis of the simulation of this PEG has shown the possibility of working in different loading ranges, both in the low frequency range up to 50 Hz and in the higher frequency range when using only the first oscillation mode up to 287 Hz. The loading mode in the region of more than 50 Hz involves the use of devices for mechanical excitation of vibrations, for example, rotary motors with magnetic carriers. Furthermore, these operating modes of the PEG can be used as the use of the PEG in the form of vibration sensors of pulse action on the structure in a certain fixed frequency range.

The considered PEG device can be upgraded and studied in more detail under other types of loading.

**Author Contributions:** I.A.P.—original draft preparation, validation of research methods; A.V.C., R.K.H.—conceptualization, review and editing, setting up experimental work, production of calculations and conclusions. All authors have read and agreed to the published version of the manuscript.

**Funding:** The work was supported by the grant No. 21-19-00423 of the Russian Science Foundation in the Southern Federal University.

**Institutional Review Board Statement:** Not applicable.

**Informed Consent Statement:** Not applicable.

**Data Availability Statement:** Data is contained within the article.

**Acknowledgments:** SFedU equipment was used. The work was supported by the grant No. 21-19-00423 Russian Science Foundation.

**Conflicts of Interest:** The authors declare no conflict of interest.

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
