# Peer review of "Parametric and Experimental Modeling of Axial-Type Piezoelectric Energy Generator with Active Base"

_applsci, doi:10.3390/app12031700_

Round 1

Reviewer 1 Report

The authors carried out the FEM simulations and experimental tests for an axial-type piezoelectric energy generator with active base. However, it seems that there are few theoretical advances and practical conclusions in this manuscript. So, it may be not suitable for publication.

  1. This generator has multiple distributed piezoelectric crystals and special mechanical constraints. Its electromechanical response is complicated. It is recommended to use spring damping model, equivalent circuit model, potential energy function and other tools for basic theoretical analysis of its electromechanical response. Otherwise, the results obtained via FEM numerical simulation may very depend on the structural parameters and have no general significance.
  2. It seems that the authors don’t take the damping effect into consideration, as no damping parameter values are shown in the parameter tables.
  3. From the simulation view, there seems to be collisions under the excitation of mechanical vibration. Do the authors consider the energy loss caused by collisions?
  4. From a practical point of view, the output energy of each piezoelectric crystal should be gathered through the circuit. The output of a single power generation crystal has no practical meaning.
  5. In my opinion, the purpose of simulation is to achieve the optimal design of the generator. However, the optimization object of this manuscript is not clear. What are the constraints of excitation, load, and generator size? The output can only be maximized through optimization under clear constraints.
  6. In some figures, the output of the generator is monotonous and has no peak. This may indicate that the range of the parameter-scanning is not appropriate.
  7. The results of simulation and experiment are not directly compared.

Author Response

The authors carried out the FEM simulations and experimental tests for an axial-type piezoelectric energy generator with active base. However, it seems that there are few theoretical advances and practical conclusions in this manuscript. So, it may be not suitable for publication.

The authors' response.  The authors thank the Reviewer for the careful reading of the manuscript and valuable quary / suggestions. The changes are highlighted with color.

Q1.This generator has multiple distributed piezoelectric crystals and special mechanical constraints. Its electromechanical response is complicated. It is recommended to use spring damping model, equivalent circuit model, potential energy function and other tools for basic theoretical analysis of its electromechanical response. Otherwise, the results obtained via FEM numerical simulation may very depend on the structural parameters and have no general significance.

The authors' response 1.
Yes. You are right, for the general formulation of the problem and the expansion of the theoretical possibilities of estimating PEG oscillations, it is possible to use simple beam models based on the theoretical approaches of Bernoulli or Timoshenko, and others, with the presence of elastic damping, including elastic elements with piezogeneration properties. There are such studies in a number of our works, as well as in the literature. They are listed in the literature. In the case of using the simulation method under any mixed conditions of coupling of various elements, a detailed study of the stress state is required. Under conditions of stress and defect concentration, the use of the finite element method is preferable. From this point of view, this approach is chosen. Our task is in this direction to build the most adequate numerical model and further study it in the field of limiting resonant loading. Due to this, a high PEG output effect is possible. The novelty of the model lies in the fact that by fixing one of the ends of the base, we forbid it to move with small deformations and shift the resonances to a conditionally large frequency range. By adjusting the form factor of the selected model, we can control the frequencies and output parameters of the model. In further research, we will try to detail this approach, including on the basis of the use of simple beam models.

Q2.It seems that the authors don’t take the damping effect into consideration, as no damping parameter values are shown in the parameter tables.

The authors' response 2.
The authors are thankful to point out the damping parameter. The damping effect is considered and the damping value is incorporated in the table 2.

Q3.From the simulation view, there seems to be collisions under the excitation of mechanical vibration. Do the authors consider the energy loss caused by collisions?

The authors' response 3.
Collisions of structural elements did not occur during experimental modeling, as well as during the excitation of mechanical vibration. The model works within the framework of elastic reciprocating deformations. The simulation results in the figures are presented on an enlarged scale.

Q4.From a practical point of view, the output energy of each piezoelectric crystal should be gathered through the circuit. The output of a single power generation crystal has no practical meaning.

The authors' response 4.
The authors are trying to do a detailed analysis of the designed system of PEG. The development of energy transmission and storage devices is not included in the purpose of the article. Piezoelectric elements can be connected in parallel, in series, or separately. The choice between the types of connection of elements depends on the device that needs to be powered: if a higher output voltage is required, then a serial connection should be selected, and if a higher output current is required, then a parallel connection.

Q5.In my opinion, the purpose of simulation is to achieve the optimal design of the generator. However, the optimization object of this manuscript is not clear. What are the constraints of excitation, load, and generator size? The output can only be maximized through optimization under clear constraints.

The authors' response 5.
The article presents the new design of Axial-type type piezoelectric generator.This design can harvest energy from d31, and d33 simultaneously. In this design, proof mass can be mounted on the duralumin beam in between piezoelectric patches and the screw side. This variation of fixing the proof mass gives flexibility in the natural frequencies of the PEGs. As per the mechanical vibration input, the first natural frequency can be adjected within the limit for high power output. This design gives flexibility and enhances the output power options compare to the other previous designs. 
The mechanical vibrations are used as an input parameter. This study is about the new design of a generator, in which proof mass has a key role to get the higher power output. The analysis is also carried out on the electrical load dependency. Further research will be carried out to improve the design of the PEC

Q6.In some figures, the output of the generator is monotonous and has no peak. This may indicate that the range of the parameter-scanning is not appropriate.

The authors' response 6.
The authors are thankful to highlight these issues. The figures are modified and incorporated into the manuscript. 

Q7.The results of simulation and experiment are not directly compared.

The authors' response 7.
The results are compared in Table 5 and Table 6.  

Reviewer 2 Report

This paper present a computational and experimental approach to modeling oscillations of a new axial- type 12 piezoelectric generator (PEG) with a proof mass and an active base. It was not well organized and there are many errors to reduce the readability of this paper. It needs to be well modified before it will be accepted.

  1. Section 4.1 introduces the principle of device operation. It had better be moved to Section 2.
  2. The novelty of this paper is not very clear.
  3. It seems that the simulation results and experimental results are not consistent.
  4. The experimental data was not enough to support this paper. For example the frequency response experiments.
  5. Why was the 10-order vibration mode simulated by the authors. Generally, it needn’t to simulate so high-order mode and it was basically useless in practical applications.
  6. Why is the captions of Figure 6 and Figure 7 totally same?
  7. In figures 6-8, the figures are not so good, The quality of these figure is poor. In figures, why not use the scientific notation or use the appropriate unit, for example, in figure 7 the unit of the displacement, where the um can be used. In figure 7 mV can be used.
  8. Figures 13 and 14 should be replotted.
  9. There are too many errors as follows:

On page 4, The caption of Table 2 is wrong.

Line 142 Page 5, Figure 4. Shows , where the dot after 4 should be deleted and Shows should modified as shows

Line 162 Page 6, In Figure 5. the first…. , where the dot after 5 should be deleted

Line 176 Page 7, the unit of 10 m/s2 is wrong.

Line 182 Page 7, the sentence of “In Figure 8 calculated …” is wrong.

In the section, why not use uW to present the output power.

Author Response

This paper present a computational and experimental approach to modeling oscillations of a new axial- type 12 piezoelectric generator (PEG) with a proof mass and an active base. It was not well organized and there are many errors to reduce the readability of this paper. It needs to be well modified before it will be accepted.

The authors' response. 
The authors thank the Reviewer for the careful reading of the manuscript and valuable quary / suggestions. The changes are highlighted with color.

Q1.Section 4.1 introduces the principle of device operation. It had better be moved to Section 2.

The authors' response 1. 
According to the authors, the best plan of the article is described as follows.
The paper is organized as follows. Section 2 Description of model parameters and electric scheme presents a description of axial-type PEG elements, a schematic description of the structure, a description of the parameters of materials used in both numerical modeling and experimental. Section 3 Modeling covers the numerical simulation of the generator. The results of modal analysis and some output data of PEG parameters as a result of harmonic modeling are presented. Section 4 Experimental probe. A description of an experimental setup for testing the operation of PEG under a certain loading is presented. A description of the results of testing PEG under dynamic loading in a certain frequency range is presented. Finally, Section 5 provides the conclusion 

Q2.The novelty of this paper is not very clear.

The authors' response 2.
The article presents the new design of Axial-type type piezoelectric generator. The novelty of the model lies in the fact that by fixing one of the ends of the base, we forbid it to move with small deformations and shift the resonances to a conditionally large frequency range. By adjusting the form factor of the selected model, we can control the frequencies and output parameters of the model. In further research, we will try to detail this approach, including on the basis of the use of simple beam models.
This design can harvest energy from d31, and d33 simultaneously. In this design, proof mass can be mounted on the duralumin beam in between piezoelectric patches and the screw side. This variation of fixing the proof mass gives flexibility in the natural frequencies of the PEGs. As per the mechanical vibration input, the first natural frequency can be adjected within the limit for high power output. This design gives flexibility and enhances the output power options compare to the other previous designs. The mechanical vibrations are used as an input parameter. This study is about the new design of a generator, in which proof mass has a key role to get the higher power output. The analysis is also carried out on the electrical load dependency.

Q3. It seems that the simulation results and experimental results are not consistent.

The authors' response 3.  
Some results of the experiment and calculation are compared in sections 4.2 4.3. According to the authors, the model has a satisfactory confirmation of the results

Q4. The experimental data was not enough to support this paper. For example the frequency response experiments.

The authors' response 4.
Some results of the experiment and calculation are compared in sections 4.2 4.3. According to the authors, the model has a satisfactory confirmation of the results

Q5. Why was the 10-order vibration mode simulated by the authors. Generally, it needn’t to simulate so high-order mode and it was basically useless in practical applications.

The authors' response 5 . 
The authors are trying to do a detailed analysis.  Corrections are incorporated in the manuscript.  

Q6. Why is the captions of Figure 6 and Figure 7 totally same?

The authors' response 6.
Corrections have been incorporated in the manuscript.   

Q7. In figures 6-8, the figures are not so good, The quality of these figure is poor. In figures, why not use the scientific notation or use the appropriate unit, for example, in figure 7 the unit of the displacement, where the um can be used. In figure 7 mV can be used.

The authors' response 7.
Corrections have been incorporated in the manuscript.   

Q8.Figures 13 and 14 should be replotted.

The authors' response 8.
Corrections have been incorporated in the manuscript.   

Q9. There are too many errors as follows:
On page 4, The caption of Table 2 is wrong.

The authors' response:  
Corrections have been incorporated in the manuscript. 

Line 142 Page 5, Figure 4. Shows , where the dot after 4 should be deleted and Shows should modified as shows

The authors' response:  
Corrections have been incorporated in the manuscript. 

Line 162 Page 6, In Figure 5. the first…. , where the dot after 5 should be deleted

The authors' response:
Corrections have been incorporated in the manuscript.  

Line 176 Page 7, the unit of 10 m/s2 is wrong.

The authors' response: 
Corrections have been incorporated in the manuscript. 

Line 182 Page 7, the sentence of “In Figure 8 calculated …” is wrong.

The authors' response: 
Corrections have been incorporated in the manuscript. 

In the section, why not use uW to present the output power.

The authors' response: 
Corrections have been incorporated in the manuscript. 

Reviewer 3 Report

  1. Which direction are the piezoelectric elements polarized, 13, 31, 33 as opposed to through the thickness?
  2. Which of your modeling elements are perfectly bonded and which of your elements have mobility to move?  Can you clarify your boundary conditions on your figure 4?
  3. How do you propose to increase the output voltage and energy density of your device?

Author Response

The authors thank the Reviewer for the careful reading of the manuscript and valuable quary / suggestions. The changes are highlighted with red color.

Q1.Which direction are the piezoelectric elements polarized, 13, 31, 33 as opposed to through the thickness?

The authors' response 1.
 The piezoelectric elements are polarized in d31 and d33 in both directions as shown in figure 2. 
Piezoelectric plates = d31
Piezoelectric Cylinder = d33

Q2. Which of your modeling elements are perfectly bonded and which of your elements have mobility to move?  Can you clarify your boundary conditions on your figure 4?

The authors' response 2.
The boundary conditions are incorporated in figure 3 ( Figure 4 old)

Q3. How do you propose to increase the output voltage and energy density of your device?

The authors' response 3.
The current design is flexible. As per the input load ( mechanical vibrations or shock), the proof mass can be added to the base plate at various locations to achieve the resonance conditions. If the resonance is achieved, the model gives higher output.  

Round 2

Reviewer 1 Report

The revised manuscript has partly addressed my concerns. However, some problems are still unsolved.

(1) I understand that the authors consider FEM simulation design as their main contribution. However, if the conclusions of simulation are not explained by theory, it may be unreliable or difficult to generalize. Therefore, I suggest that the authors do some theoretical analysis from the view of the potential energy function.

(2) I understand that the authors do not want to be involve in the energy harvesting circuit at this stage. But do the current results realize the maximum of the total power from all piezo blocks, or just the maximum of one piezo block’s power?

(3) My previous concern 5 is still not addressed. What is the optimal problem? What’s its object and constraints?

(4) I suggest the authors compare their works with the most similar literatures, including their own published papers, and better summarize their novel contributions.

Ref:

Haldkar R K, Parinov I A, Cherpakov A V, et al. Modelling vibrations of axial piezoelectric generator with active base[C]//Journal of Physics: Conference Series. IOP Publishing, 2021, 2131(2): 022018.

Zhou K, Dai H L, Abdelkefi A, et al. Theoretical modeling and nonlinear analysis of piezoelectric energy harvesters with different stoppers[J]. International Journal of Mechanical Sciences, 2020, 166: 105233.

Huang X, Zhang C, Dai K. A Multi-Mode Broadband Vibration Energy Harvester Composed of Symmetrically Distributed U-Shaped Cantilever Beams[J]. Micromachines, 2021, 12(2): 203.

Liu H, Lee C, Kobayashi T, et al. Investigation of a MEMS piezoelectric energy harvester system with a frequency-widened-bandwidth mechanism introduced by mechanical stoppers[J]. Smart Materials and Structures, 2012, 21(3): 035005.

Li X, Yu K, Upadrashta D, et al. Multi-branch sandwich piezoelectric energy harvester: Mathematical modeling and validation[J]. Smart Materials and Structures, 2019, 28(3): 035010.

If the second revised manuscript still does not meet the requirements for publication, I will have to recommend rejection of the manuscript

Author Response

(1) I understand that the authors consider FEM simulation design as their main contribution. However, if the conclusions of simulation are not explained by theory, it may be unreliable or difficult to generalize. Therefore, I suggest that the authors do some theoretical analysis from the view of the potential energy function.

authors response:

As part of these studies, we presented an approach based on the application of the finite element method. The theoretical elements of the FE method are given in the text as a separate chapter. Analysis, based on the solution of a number of direct problems of modal and harmonic analysis showed satisfactory agreement with experiment. We are grateful to the referee for advice and in the future we will approach research based on the use of analytical modeling, taking into account elements of potential energy functions.

(2) I understand that the authors do not want to be involve in the energy harvesting circuit at this stage. But do the current results realize the maximum of the total power from all piezo blocks, or just the maximum of one piezo block’s power?

authors response:

Within the framework of these studies, it is of interest to collect information about the operation of each piezo element separately. In the description of literature sources, we separately included a link to various schemes for connecting the load to the piezoelectric elements of the generator. Source number [18].

(3) My previous concern 5 is still not addressed. What is the optimal problem? What’s its object and constraints?

The authors thank the reviewer for their attention. Yes. We really missed this moment. Reviewer's comments resolved. Tex is marked in blue in the modeling chapter.

authors response:

(4) I suggest the authors compare their works with the most similar literatures, including their own published papers, and better summarize their novel contributions.

authors response:

At the suggestion of the reviewer, corrections were made to the text of the first chapter. Updated review and literature sources. We have tried to better structure the description of the problem. Elements of the updated text are marked in blue.

Reviewer 2 Report

All the concerns have been addressed by the authors. Now it can be accepted.

Author Response

The authors thank the reviewer for their attention
